# The Journey to Early Identification and Intervention for Children with Disabilities in Fiji

**DOI:** 10.3390/ijerph20186732

**Published:** 2023-09-07

**Authors:** Fleur Smith, Sureni Perera, Manjula Marella

**Affiliations:** 1Nossal Institute for Global Health, Melbourne School of Population and Global Health, The University of Melbourne, Melbourne 3010, Australia; marella.m@unimelb.edu.au; 2Head Office, Frank Hilton Organization, Lot 139 Brown Street, Suva, Fiji; sureni@hilton.org.fj

**Keywords:** children with disability, developmental disability, early childhood intervention, Fiji, low- and middle-income country (LMIC), journey mapping, caregivers, coordination of care

## Abstract

Early identification of developmental delay or disability and access to early intervention improves outcomes for children with disabilities and their families. However, in many low- and middle-income countries, services and systems to enable timely, co-ordinated care and support are lacking. The aim of this research was to explore the experiences of families of children with developmental disabilities in Fiji in accessing services for intervention and support across sectors. This qualitative study involved conducting interviews with caregivers of children with disabilities *(n =* 12), and relevant key stakeholders from health, education, disability, and social support sectors (*n =* 17). We used journey maps to identify key stages of the families’ journeys, identify key barriers and enablers at each stage, and provide multi-sectoral recommendations for each stage. Enablers include proactive help seeking behaviours, the use of informal support networks and an increasingly supportive policy environment. Barriers to identification include a lack of awareness of developmental disabilities and the benefits of early intervention among service providers and the community. A lack of service availability and capacity, workforce issues, family financial constraints and a lack of collaboration between sectors were barriers to intervention once needs were identified, resulting in significant unmet needs and impacting inclusion and participation for children with disabilities. Overcoming these challenges requires a multi-sectoral approach.

## 1. Introduction

The earlier children with, or at risk of, developmental disabilities are identified and receive timely, coordinated support and intervention, the better their outcomes and those of their family are likely to be [1,2]. However, implementing effective systems for early identification, intervention and support for young children with developmental delays and disabilities is complex, and little evidence exists on effective practices for coordinated care across sectors. The provision of care and support for children with disabilities typically involves multiple sectors including health, education, and social support services, but multi-sectoral collaboration to ensure coordinated, comprehensive support is frequently lacking [3]. This creates challenges for families navigating services and results in children with developmental disabilities ‘falling through the cracks’ of service and support systems.

Services for children with disabilities in many low- and middle-income countries (LMICs) including Fiji are limited. Poor awareness of developmental disabilities, insufficient funding, lack of workforce, fragmented referral systems and an absence of contextually appropriate tools for early identification of disabilities have been identified as barriers to early childhood intervention (ECI) program access and growth [4]. Those services that exist are typically provided by civil society organisations or within a segregated system [5,6,7] rather than well-integrated into government systems. Regardless of the sector or whether services are provided by government or non-government organizations, adequate governance, guidance and coordination mechanisms remain necessary to ensure children with disabilities and families receive appropriate supports early and when they need them [8].

The Sustainable Development Goals, adopted by all United Nations member states in 2015, aim to end poverty and inequality, protect the planet and create the conditions for sustainable, inclusive economic growth and shared prosperity [9]. The specific inclusion of early childhood development in Sustainable Development Goal target 4.2 (access for all children to quality early childhood development, care and pre-primary education) [9] acknowledges the importance of the crucial early childhood years, and recognizes there are consequential advantages for society when all children have the opportunity for optimal development. For children with disabilities to realize their potential alongside their peers, evidence suggests that an inclusive ECD approach must include early identification of developmental disabilities, inclusive universal services and specific early childhood intervention (ECI) [4,7,10].

The Nurturing Care Framework for Early Childhood Development (NCF) developed by the World Health Organization (WHO), UNICEF and the World Bank recognises that supporting children’s development requires the collaboration and coordination of multiple sectors [11]. This may be even more critical for children with developmental disabilities, who often require an even greater range of supports spanning sectors that include health, education and social services. Given the high prevalence of developmental disabilities in LMICs [12], greater attention is needed to ensure multisectoral ECD programming in these contexts is inclusive of the specific needs of children with developmental disabilities.

Pacific Island countries have recently established the Pacific Regional Council for Early Childhood Development (PRC4ECD), a multi-sector, multi-government body, to guide and strengthen approaches to ECD across the Pacific [13] and implement the Pasifika Call to Action on ECD [14]. The Nurturing Care Framework [11] and the Pasifika Call to Action both promote multi-sectoral collaboration and coordination to enable the best possible outcomes for young children, and this must include those with or at risk of developmental disabilities. A recent WHO Global report on health equity for persons with disabilities also highlights the importance of investing in models of care that focus on ECD for children with disabilities and advocates for cross-sector collaboration to achieve this [15].

### Fijian Context

Fiji is a middle-income Pacific Island nation and a regional leader in the Pacific with established health, education, and social sectors. With a total population of 884,887, approximately 92,000 are children under 5 years [16].

The Fijian archipelago consists of 332 islands, although 70% of the population live on the main island of Viti Levu and are concentrated around the capital city, Suva. For the 44% of the population living in rural and remote island areas, travel to and from the larger urban centres where most services are located is challenging, which impacts access to specialist health care and other supports. The use of traditional medicine and other cultural practices remain common.

In the 2017 census, 13.7% of the population reported at least one area of functioning difficulty [16]. Preliminary results from the 2021 Multiple Indicator Cluster Survey (MICS) indicate that 8.8% of children aged 2–17 years in Fiji were reported to have functioning difficulty in at least one domain, and just 82.9% of children 36–59 months were considered to be ‘developmentally on track’ [17].

Services for children with disabilities have existed in Fiji for some time, but have been primarily for school-aged children through special schools. The Frank Hilton Organization (FHO, formerly The Fiji Crippled Children’s Society—Suva Branch) is a non-profit organisation established in the 1960s to provide care and education for children with disabilities. FHO have established an Early Intervention Centre that provides individual- and group-based services for young children with disabilities and their families [18]. Other services for children with developmental disabilities in Fiji are limited but include a Developmental Paediatric service at Colonial War Memorial Hospital (CWM) in Suva, hospital-based physiotherapy services, special schools, a diminishing community rehabilitation assistant program and NGO-run vision and hearing services. Rehabilitation services within the Ministry of Health and Medical Services primarily cater for adults. Fiji lacks an allied health workforce including occupational therapists and speech pathologists and local training programs to facilitate workforce development.

Fiji has ratified the United Nations Convention on the Rights of Persons with Disabilities (CRPD) [19] and Convention on the Rights of the Child (CRC) [20], both of which outline specific obligations to children with disabilities. At a national policy level, in 2018, Fiji adopted a national Rights of Persons with Disabilities Act and has policies on special and inclusive education that include early detection, intervention and education. Fiji is currently drafting its first ECD policy, the National Persons with Disability policy is undergoing review, and is soon to release a National Disability Inclusive Health and Rehabilitation Action Plan.

Identifying children with, or at risk of, disabilities early and connecting them and their families with appropriate support is vital for improving outcomes and removing inequities. To do so, we first need to understand current practices and pathways taken by families of children with disabilities when attempting to access support. This study aimed to understand the demand and supply-side factors influencing access to early identification and intervention for children with disabilities in Fiji. We examined the experiences of families of children with disabilities in accessing services for intervention and support and sought information on current practices from key sector stakeholders involved in services for children with disabilities. We used journey maps to identify the key stages of the families’ journeys, identify key barriers and enablers at each stage, and provide multi-sectoral recommendations for each stage.

## 2. Materials and Methods

A qualitative study utilizing in-depth interviews with caregivers of children with disabilities and key stakeholders from relevant sectors was conducted to explore both supply-side and demand-side factors influencing identification of developmental disabilities, and the access and uptake of services for young children with disabilities.

The study was conducted between February and June 2021 in 3 areas of Fiji—Suva (urban), Serua (rural) and Kadavu (remote island)—to capture the perspectives from those in diverse communities. An advisory group was established at the commencement of the study to provide advice and oversight of contextual issues relevant to the study design, participant recruitment, validation of findings and review of study deliverables. Advisory group members included government and non-government service providers, representatives of key ministries, and caregivers of children with disabilities. Due to the travel restrictions associated with the COVID-19 pandemic, data collection was conducted jointly by the Melbourne-based researcher (FS) and a local Fiji-based researcher (SP). FS conducted interviews with key stakeholders remotely using Zoom. SP conducted face to face interviews with caregivers in all three areas, and with stakeholders in Serua and Kadavu where there was insufficient internet connection for remote interviews.

Ethics approval was obtained from the Human Research Ethics Committee of the University of Melbourne and the Fiji National Research Ethics Review Committee at the Ministry of Health and Medical Services.

### 2.1. Participants

A total of 12 caregivers participated in the study, with four caregivers recruited from each study location through existing networks. Caregivers of children with a range of conditions or impairments and of varying ages were purposively selected from FHO and other service provider networks to capture experiences of the journey of accessing care at different developmental stages, and included those at various stages of accessing intervention and supports. The characteristics of caregiver participants are further outlined in Table 1, and child characteristics in Table 2.

Caregivers were interviewed using a semi-structured interview guide and were asked about their experience of identification or assessment of disabilities, referral pathways and access to services and supports, impact of supports (or lack thereof) on the child and caregiver, their perceptions of disabilities and intervention, and factors impacting care/support seeking behaviours in relation to disabilities. Language translation was provided to enable caregivers across ethnic groups to participate.

Key stakeholder interviews were conducted with representatives of government and service provider levels from the health (*n* = 7), education (*n* = 3), social support (*n* = 4), and disability sectors (*n* = 3). Of the 17 stakeholder interviews, 16 were individual interviews and one group interview, with a total of 19 participants. The majority of participants were female (*n* = 15). Stakeholders were identified by the advisory group and invited via email or phone to participate. Some additional participants were identified by snowball sampling during the fieldwork. The characteristics of key stakeholder interviews are further outlined in Table 3.

Key stakeholders were interviewed using a semi-structured interview guide and were asked about their knowledge, perceptions and experiences of current practices in early identification and intervention for children with developmental disabilities, including factors impacting on referral, access, uptake and coordination of care for children with, or at risk, of disabilities.

All interviews were audio recorded and professionally transcribed and translated into English where interviews were conducted in the local language. Translated transcriptions were validated by the Fiji research team.

### 2.2. Data Analysis

Transcriptions of interview recordings were entered into the coding software NVivo Release 1.7.1 (Lumivero, Denver, CO, USA) for analysis by FS and MM. Thematic data analysis was conducted inductively and deductively. FS and MM coded data independently and compared. Where there were discrepancies, these were discussed and resolved mutually. The objectives of the research provided an initial structure for thematic analysis, with emerging themes coded drawing on the ‘enabling environments for nurturing care’ from the NCF [11].

The 12 caregiver interviews were analysed to map the steps each family had undertaken from birth to the time of the interview to identify their child’s needs, access intervention, and the barriers and facilitators encountered along the journey. FS performed the initial mapping and these were reviewed by MM and SP. The journey maps were then further analysed to identify common experiences and were synthesised into 3 composite journey maps that reflected the broad experience of participants. Barriers and facilitators identified in the thematic analysis were also mapped to the key journey stages.

Journey mapping is adapted from the market research sector and is a process of examining individual’s stories to understand their unique and complex experiences of accessing services and systems [21]. Journey maps aid in identifying gaps, barriers and facilitators in service systems by mapping a series of ‘touchpoints’ or interactions between the ‘consumer’ and the ‘service system’ [22]. It is a relatively new approach being used in health and medical research to examine the patient health care experience and determine opportunities for changes in delivery of care [23]. Examples are emerging of journey mapping being used in the disability and social support sector, including as part of a program evaluation in Minnesota to examine the journey of families of children with autism as they navigate through multiple support systems [24].

Findings were presented at a workshop with the advisory group for discussion and feedback, and input obtained for formulating recommended actions for the local context.

## 3. Results

### 3.1. Journey Mapping

Analysis of the 12 maps demonstrated three key stages experienced by families of children with disabilities, each with their own barriers and facilitators, and opportunities for strengthening. These stages are shown in Figure 1.

After mapping and analysing the journeys described in the 12 caregiver interviews, three common pathways emerged that were synthesised into three composite journeys—(1) children with risk factors or developmental conditions identified at birth; (2) children without any risk factors/developmental conditions identified at birth; and (3) children from remote island communities. These composite journeys are shown in Figure 2, Figure 3 and Figure 4 and highlight some commonly reported experiences.

Each composite journey map shows the experiences of a fictional child and their family from birth to the time of the interviews. By considering the experiences at the different stages along the journey, we documented barriers and enablers at these different time points, which enabled the identification of actions to improve the experience.

### 3.2. Thematic Analysis

Themes from both caregiver and stakeholder interviews were mapped to the three journey stages and explored in terms of barriers and facilitators to timely and appropriate intervention and support. A further overarching theme of ‘cross-sector collaboration’ was also explored, which identified systemic factors impacting across all stages of the journey.

#### 3.2.1. Stage 1: Identification of Needs

##### Missed Opportunities for Identification

In stage 1, most service interactions were with the health sector. This included hospital-based birth and neonatal services, medical follow-up for identified complications or congenital conditions, and routine health care for illness or maternal and child health services, known as ‘baby clinics’ in Fiji. Findings suggest the focus of these interactions was on the child’s medical and health needs with little in the way of discussion about the child’s development, or identification, referral or monitoring of known or potential developmental concerns by health workers.

Most medical follow-up appointments occurred at Fiji’s main tertiary hospital in the capital, Suva. While most caregivers expressed an intention to attend these appointments, distance to services and cost of transport was a barrier, especially for those from rural and remote areas. This led to lost continuity of care and opportunities for monitoring and identification of developmental concerns.


*Sometimes even cases who are booked don’t come…we always call our patients a day prior to the clinic. And they say that they’re coming, but because of some bad weather conditions or transportation problems, financial issues, they don’t turn up.*
(Stakeholder, Health)

All caregivers reported attending routine ‘baby clinics’ with their child. Both caregivers and stakeholders described ‘baby clinic’ visits as being a widely accepted and adhered to practice in Fiji, with these a key first point of contact to monitor and detect developmental delays or disabilities. However, caregivers reported their child’s development was not discussed at these clinics. Record cards completed by nurses during baby clinics include items for developmental milestones, but caregivers reported that delayed milestones were not noted on the card, with some caregivers providing examples of milestones being recorded as met many years before they were attained.

Beyond the health sector, it was reported that children with disabilities may be identified by social-welfare outreach visits, or by teachers once attending education settings at 6 years of age, missing the early years. However, children with mild developmental concerns were less likely to be identified by teachers or social welfare officers.

To improve early identification, it was noted that the CWM Paediatric Department is working to build the capacity of health workers in identifying developmental disabilities and routine screening of all children presenting to hospital services.

##### Lack of Information for Caregivers

Caregivers of children with known developmental conditions or risk factors from birth reported a lack of information or communication from health care staff about their child’s condition or any potential implications for their child’s development. Some reported being told of a diagnosis but of not receiving any other information about what this meant for their child’s development, additional supports that their child may need, or advice for managing their child’s needs. Others reported that they were not told of their child’s condition at all.


*Parent: They just told me Down Syndrome without even discussing the meaning of this word.*



*Interviewer: were you given any advice about how to care for your child?*



*Parent: I wouldn’t say they gave me advice, the doctor only said they need to take the blood. I asked what are the bloods for and she said that’s because she [baby] has Down Syndrome. That’s it. Just that.*
(Caregiver, Serua)

Health sector stakeholders acknowledged that the focus on health needs during health care interactions was a barrier to the identification of developmental disabilities and referrals for early intervention.


*We normally address care on a child as general, as a child health without the focus on whether they have disability or not. They get sick, they come and see us, and it’s just when they’re sick then we intervene. But there are other special areas of their needs that are being ignored at the moment which we think we need to improve on.*
(Stakeholder, Health)

Stakeholders attributed this lack of communication by health workers to poor knowledge and awareness of child development and disabilities, the benefits of early intervention, or the available services for referral and support. Some caregivers reported that they may have sought help sooner if they had information about their child’s condition earlier.


*Maybe if we were familiar with the symptoms at the very earlier age maybe we could have gone [to intervention] early. From hospital, we could have prepared to come early [to early intervention service], then maybe we could have, you know tackle her situation a bit more, because I kept thinking she’s going to grow out of it you know. But at that time we just left it and we didn’t ask for help’.*
(Caregiver, Suva)

##### Emerging Caregiver Concerns and Help-Seeking

All caregivers reported an early awareness that their child’s development was different to that of other children, with many noticing that their child was not moving (e.g., rolling over, crawling, standing) or talking at the same age as other children. They reported either not seeking help, believing their child to be ‘just slow’, or sought help during health service interactions and had their concerns dismissed. Several caregivers said they were advised ‘to wait and see’ or that their child was ‘just slower than other children’. Consequently, many did not pursue further assessment and intervention until many months or years later, if at all.


*They just told me that it will take him a long time to do something. When I came [to the clinic], at 1 year he was still only rolling around. Then they told me, it’s normal, Down Syndrome, they are weak. After that I didn’t take him anywhere, we just stayed together, and I’ve been taking care of him.*
(Caregiver, Serua)

However, some caregivers described proactively seeking information and support despite concerns being dismissed, with the internet or family members reported as common sources of information and support.

While stakeholders acknowledged the challenges with delayed identification and support from the health care system, they also highlighted challenges with delayed help seeking and presentation of children with development delays or disabilities to services. Various barriers were reported by the stakeholders including negative prior health care experiences, prevailing community stigma surrounding disabilities, and cultural norms of respect for authority resulting in caregivers not questioning health workers when their concerns were dismissed.


*One of the main cause [of parents not seeking help] is stigma. And the fear of knowing, you know, stigma in the sense. If they feel that their child is developmentally slow, they will usually keep that child at home, and not seek attention, because that would bring out the fault within the kid and in the family. And fear of knowing, because if they do accept that something is wrong, then there is something is wrong.*
(Stakeholder, Health)

However, community rehabilitation assistants (a cadre of community-based nurses with rehabilitation training) described that it was not uncommon for caregivers to approach them when they are out in the community with concerns about their child, indicating the potential benefits of community-based programs for increasing awareness and help-seeking.

##### Recommended Actions Targeting Stage 1—Identification of Needs

Training on early identification of disabilities and referral for those who are the first point of contact for families. This may include health workers at the primary care level (e.g., village nurses/community health workers), teachers and early childhood educators, and welfare officers.Strengthen processes for developmental monitoring at routine baby clinics.Improve information given by health workers to new parents about their child’s condition, potential developmental implications, and sources of support available.Consider mechanisms for financial support for families of children born with complications to attend follow-up health and medical appointments to enable continuity of care and not being ‘lost’ to follow-up.Raise community awareness of developmental disabilities, the benefits of timely early intervention, and the services available.

#### 3.2.2. Stage 2: Accessing Intervention and Supports

In stage 2, families were starting to become aware of and utilise intervention and support services. Barriers and facilitators to access were identified both at the family and community level, and at the service and system level.

##### Barriers to Accessing Services

Caregivers and stakeholders agreed there is a lack of services to provide early childhood intervention and other supports for young children with developmental disabilities and their families in Fiji. This is particularly the case outside the major urban areas.


*If children are identified in Lautoka they have to come all the way to Suva. If our children in the islands need services, they have to wait till someone can pay for them coming…this is one of the areas that we’re struggling with.*
(Stakeholder, Social Welfare)

Where services exist, caregivers do not know about them or lack awareness of the value of early intervention. Some caregivers reported believing intervention before 4 or 5 years of age to be a ‘waste of time’ and that children should be given time ‘to catch up’. Stakeholders reported many in the community believe special schools are the earliest and main support available for children with disabilities and do not consider accessing services prior to school age.

The distance and cost of transport to services are significant barriers for families, especially those in rural and remote areas where transport services are limited and access is impacted by changing tides and weather. While some services report providing outreach services to rural and remote areas, these are currently limited to only some parts of Fiji.


*So, fare is the main problem and secondly he is in diaper, I have to buy his diaper and other expenses to travel to town, and we have to buy something to eat.*
(Caregiver, Serua)

Competing priorities were also reported by caregivers as a barrier to utilising existing services, reporting difficulties attending appointments due to work and family commitments or juggling multiple appointments for their child.

Extensive waiting lists are reported due to a lack of service capacity. Among stakeholders, there is a perceived lack of priority by government to resource early childhood intervention services and other supports for children with disabilities, with services reporting they must rely on donor funding and fundraising to supplement government funding. Lack of an adequately trained workforce to provide interventions, and a lack of funded positions and career development opportunities for those seeking to work in ECI and other disability services were also reported.

##### Facilitators of Access to Services

Despite the barriers, caregivers reported proactively seeking support for their child, utilising informal support networks, and developing their own strategies to address support needs. Awareness of disabilities generally in the community was reported to be improving, largely via word of mouth and information sharing via social media, supported by recent policy and legislation that promotes disability inclusion. Stakeholders felt this was leading to more acceptance and awareness of supports by caregivers, and a greater understanding of the rights of children with disabilities.


*The awareness is there now, people now are coming out, like the parents they used to hesitate to come to us and say, ‘My child is blind’. But with lots of awareness and counselling, they have now come to us for support.*
(Stakeholder, Disability)

Relationship-building between service providers was reported to be facilitating greater awareness of services and improving referral pathways, while building relationships at the community level, including through engaging village leaders, was identified as important in building trust and acceptance of services.

##### Recommended Actions Targeting Stage 2—Accessing Intervention and Supports

Financial support mechanisms for caregivers to take time off work to attend appointments for intervention.Transport allowances or subsidies for travel to and from services for families of children with disabilities.Onward referral to local services where available for follow-up of rural and remote children (e.g., hospital-based physiotherapy services).Funding for new and existing services to improve capacity and coverage outside the main urban areas.Extend the use of outreach and remote/telehealth models of service delivery drawing on learnings from existing community-based programs.Strengthen existing national policies and action plans on ECD and disabilities to explicitly include children with disabilities, backed with appropriate resourcing.Develop locally contextualized, sustainable mechanisms to build the capacity of the early intervention/disability support workforce that include training, mentoring, and opportunities for career progression.

#### 3.2.3. Stage 3: Outcomes—Impact of Intervention and Unmet Needs

In stage 3, caregivers described the impact that access, or lack thereof, to intervention and supports has had on their child and family, and ongoing unmet support needs.

##### Benefits of Access to Intervention and Supports

For children who received intervention, caregivers reported improved function, independence and participation in family and community life. Caregivers also described benefits for themselves, including having a better understanding of their child’s condition and strategies to promote their abilities and inclusion, and feeling empowered to care for, support and advocate for their child.


*They [early intervention service] are able to tell us some of the techniques to do at home to help [my child]. Now he can sit on a chair together with the [other] students, and one of his biggest achievements now is that he is able to get down from the bed and go up again and get down by himself.*
(Caregiver, Suva)

Caregivers also reported highly valuing the peer support they received from meeting other parents of children with disabilities when attending services.

##### Unmet Needs and Impact of Lack of Access to Intervention and Supports

Many children with disabilities and their families were reported to have significant unmet support needs. Negative impacts relating to developmental skills and functioning, access to the community, access to education, ongoing stigma and exclusion, and caregiver support were all discussed as negative outcomes of a lack of timely access to intervention and supports.

Caregivers reported that without appropriate support and intervention, their child missed out on opportunities to optimise independence and function, and are excluded from many aspects of daily life, such as playing with peers, attending school and engaging in community activities. Caregivers also reported feeling unsupported, isolated and lacking strategies to interact with and support their child.


*Sometimes she wants something and I don’t really know what is it. She’ll be pointing and doing all this and indicating this and I’ll just say, “what is it you want? You tell Mama. Tell Mama, come on, you can do it.” And she’ll just start pointing. That’s the biggest challenge, the communication.*
(Caregiver, Suva)

According to stakeholders, while some organisations of persons with disabilities (OPDs) in Fiji provide supports for children and FHO provides advocacy and caregiver support, there are no OPDs specifically for children with disabilities and their families.

The lack of availability of paediatric assistive technology (AT) was also reported as an issue with wide-ranging implications for the dignity, safety, and wellbeing of children and caregivers, and impacting access, inclusion, and participation.


*I need one wheelchair…I just want him to get something to make him go here and there. Like even when he wants to go to town or something like that. I have to carry him and because I have 3 children, so I have to look after [all of] them.*
(Caregiver, Suva)

Stemming from a lack of support to promote function and independence, many caregivers reported a reluctance to send their children to school. They cited a lack of communication skills, not being toilet trained, and mobility issues as reasons for their children not attending school or early education. Additional barriers reported were a lack of awareness of their child’s right to attend school and the policy requirement in Fiji of education settings to be inclusive.

Further, families of children with disabilities reported ongoing financial pressures. While most caregivers reported receiving the disability allowance, several described having to choose between meeting everyday household needs and disability-specific costs for their child. In some cases, it was reported that siblings miss school to care for children with disabilities so that parents can work. Caregiving pressures are compounded by a lack of properly resourced disability-inclusive early childhood education centres and schools.

##### Recommended Actions Targeting Stage 3—Outcomes—Impact of Intervention and Unmet Needs

Develop and resource models of care to provide interim advice and support for families on service waiting lists.Establish an OPD or advocacy group specifically for children with disabilities and their families, providing representation and voice to the lived experience of families of children with disabilities.Establish parent self-help groups, with consideration of options to meet in person or virtually using phone or digital technology, if available, to overcome issues of distance.Review of the disability allowance to reduce financial pressure on families of children with disabilities.Promote community awareness of the right to education and early education for children with disabilities, alongside adequate resourcing for inclusion in mainstream education settings.Increase resourcing for disability-inclusive ECE/childcare to enable caregivers to work and opportunities for appropriate stimulation and early learning for young children with disabilities.Establish mechanisms for the funding, procurement, and supply of paediatric assistive technology including options for local fabrication and maintenance.Establish a pool for paediatric equipment that children have outgrown to be re-issued to other children.

#### 3.2.4. Cross-Sector Collaboration

Stakeholders were asked their views on collaboration between sectors involved in the support of children with disabilities and their families. Both positive experiences and opportunities for strengthening were identified.

Stakeholders reported that a lack of clear communication and co-ordination mechanisms between sectors impacts the timely identification, referral, and holistic support of young children with disabilities and their families. It was described that each sector currently has its own tools and mechanisms for identifying disabilities, largely to determine eligibility for the supports they provide. However, even when referrals are made across services or sectors information is not routinely shared in a way that facilitates coordinated care. One health sector stakeholder gave the example of children being referred to health services from the education system, but relevant information such as reason for referral was not shared.

Several stakeholders identified a lack of clarity or agreement at a government level of who is responsible for what with regard to early intervention, and children with disabilities more generally. Early intervention was described by some as appearing to ‘fall through the cracks’ with ‘a lot of checking boxes but no one actually taking responsibility’.


*If it’s like funding for children, then Ministry of Education, go okay that’s our funding because it’s children. But they don’t talk to health to say, okay, you do the health aspect and we do the education aspects,….. so, if there is better coordination, then the resources can be better put to use.*
(Stakeholder, Disability)

Stakeholders discussed a need for clarification of shared responsibilities, improved mechanisms for communication, and formalised systems for referral and sharing of information.

Positive examples were reported of collaboration beginning through emerging networks and referral pathways between and within organisations, with these largely driven by motivated individuals. Further, at a policy level, Fiji was reported to have a growing focus and commitment to both early childhood development (ECD) and disabilities more broadly. Stakeholders felt these policies and strategic frameworks can be further strengthened by explicitly being inclusive of young children with disabilities, and these must be supported by plans for systemic change.

##### Recommended Actions Targeting Cross-Sector Collaboration

Key ministries to have a disability focal point who collaborate with each other to coordinate actions that support for children with disabilities.Establishment of formal agreements and guidelines between health, education, social welfare and disability sectors as to responsibilities with regard to children with disabilities.Establish effective referral mechanisms between service providers across sectors and appropriate sharing of information while ensuring privacy and confidentiality.Consider developing a common identification tool and referral form, using consistent language across sectors.Ensure introduction of any identification and referral mechanisms is supported by training for relevant workers in each of the sectors.Develop a directory of services available for children with disabilities that is available for all sectors and the community.

## 4. Discussion

This study used journey mapping to examine the pathways undertaken by families of children with disabilities in Fiji to identify disabilities and access intervention and support for their children. The journey-mapping process enabled examination of the various system touchpoints encountered by families from the child’s birth onwards, and factors that inhibited or facilitated successful and timely access to services and supports at different stages of the journey. The journey maps also provided a means to compare journeys and highlight factors that were commonly experienced, and triangulate information from stakeholders. Recommended actions were then presented for the consideration of actors across sectors in Fiji, with some relevant for one sector or ministry, and others requiring collaboration across sectors.

Ly et al. [23] explored the use of journey mapping in palliative care, another area of care provision where patients or ‘consumers’ may encounter multiple service providers and sectors. They argue that journey-mapping provides a person-centred perspective of the lived experience of a service system, is effective in identifying barriers and enablers in care provision and provides opportunity to proactively address these. Similarly, Underwood et al., in a study of mapping tools for use in evaluating early childhood services, argue that journey maps are a useful method to understand the ‘service user’ perspective to better support service access [21].

Utilizing the service user perspective, our study found that families have many touchpoints with the health sector in the first years of a child’s life. Building the knowledge and capacity of health workers and other potential first points of contact, such as early childhood educators, to identify developmental delays and disabilities is key to building effective systems for early intervention. However, outcomes will not be improved without also increasing the availability and capacity of services to provide intervention and support, particularly outside main urban centres.

Our study identified systemic issues of lack of funding and prioritization by government, and a lack of workforce and training opportunities as contributing to the limited growth and capacity of both universal services and disability-specific services in Fiji to respond to the needs of children with disabilities. Addressing these systemic issues will require the relevant sectors to work together. Pleasingly, there are already examples in Fiji of service providers across sectors working to improve referral pathways, which may be supported and strengthened by collaboration at the ministry levels to establish formal agreements and guidelines on service provision for children with disabilities.

While the barriers to early identification and intervention identified in this study are not unique to Fiji, but common to many LMIC contexts and well documented [5,6,8], this research highlights challenges and potential opportunities to address them at specific touchpoints of the multi-sector service and support systems. The caregiver perspective of navigating services when combined with information from key stakeholders made it possible to identify practical actions to improve care pathways and outcomes for children with disabilities. This study supports current frameworks such as the NCF and Pasifika call to Action calling for a multi-sector approach to inclusive ECD. In the Fiji context, findings from this study could be used to ensure that the needs of children with disabilities are adequately represented in the forthcoming ECD and disability policies, and rehabilitation action plan.

There are limitations to be considered for this study. This study included a small sample of caregivers from only three areas of Fiji, hence the findings provide just a small insight into the journeys of families of children with disabilities and may not represent the experiences of families of children with disabilities more broadly in Fiji. Given the sample of caregivers were purposively identified, the sample could reflect families with better knowledge about their children’s needs and available services. Further evidence is needed on families who have missed opportunities to access care.

## 5. Conclusions

Opportunities are being missed for the early identification and referral of children with developmental delays and disabilities in Fiji. Once needs are identified, lack of services and workforce, ad hoc referral systems, and family financial constraints lead to unmet support needs, impacting child and family wellbeing. Addressing these issues needs a cross-sector approach. Journey-mapping has been a useful method for understanding barriers and enablers experienced by families of children with disabilities across systems at different stages of their journey, and to identify targeted actions to strengthen systems and improve outcomes.

## Figures and Tables

**Figure 1 ijerph-20-06732-f001:**
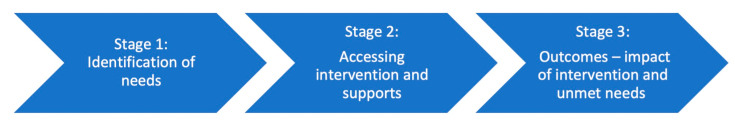
Key journey stages.

**Figure 2 ijerph-20-06732-f002:**
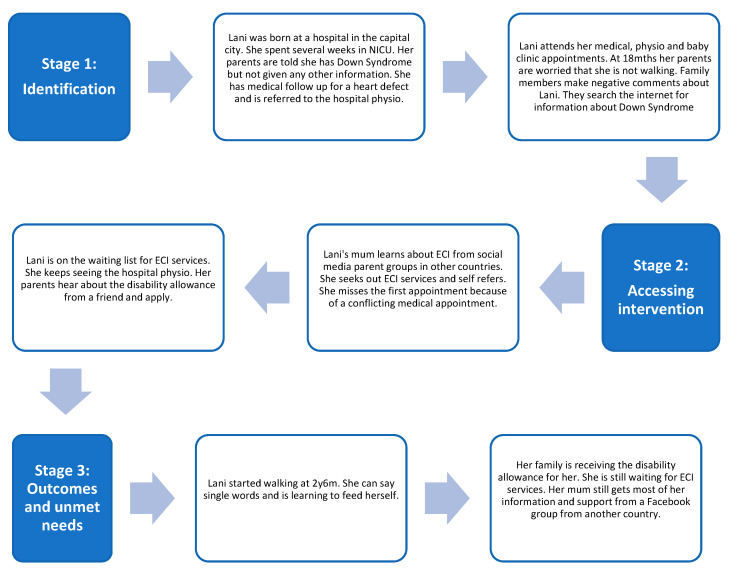
Journey of a child with risk factors/developmental condition identified at birth. Lani is a 3-year-old girl who lives with her parents and baby brother in Suva, the capital city. Lani has Down Syndrome. She started walking about 6 months ago and she is saying single words and using gesture to communicate. She can feed herself and is starting to learn toilet training. She loves playing with her little brother.

**Figure 3 ijerph-20-06732-f003:**
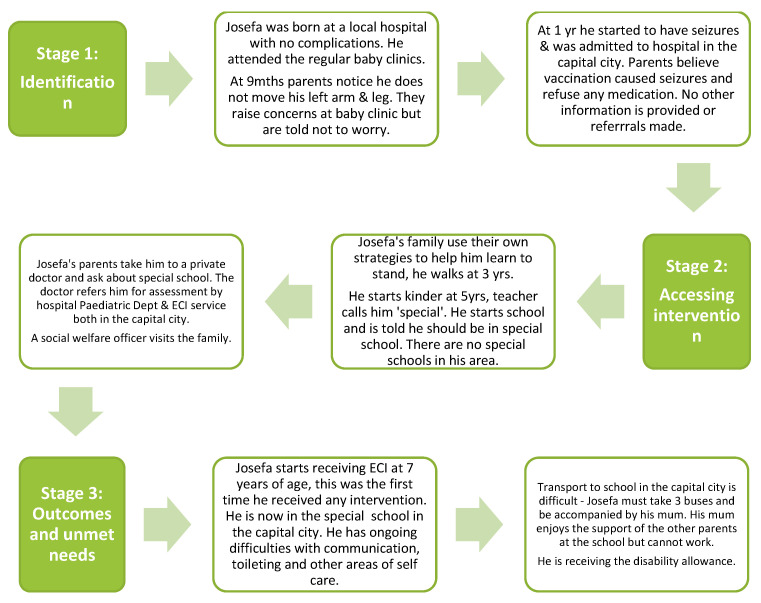
Journey of a child without any risk factors/developmental conditions identified at birth. Josefa is a 12-year-old boy living in Serua (rural area). He is the eldest of 5 siblings. He has cerebral palsy and epilepsy. He has weakness in his left arm and leg and his speech is difficult to understand. He is not yet fully continent and needs help with getting dressed. He likes to play with the soccer ball.

**Figure 4 ijerph-20-06732-f004:**
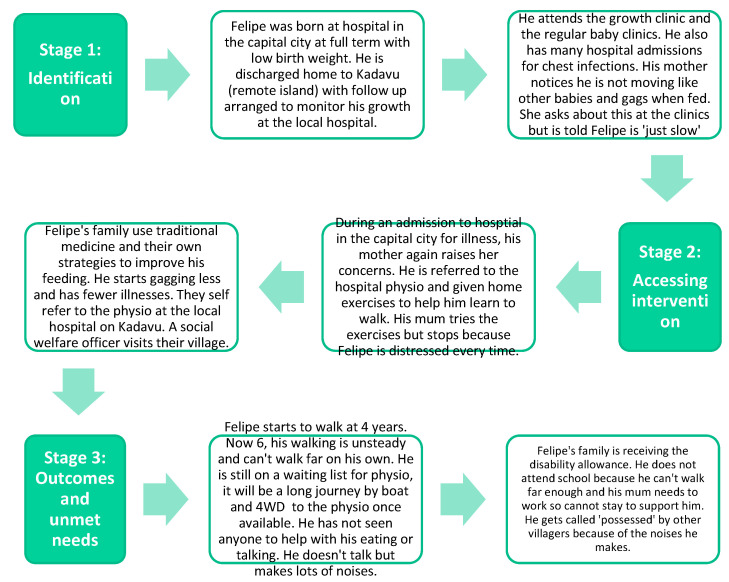
Journey of a child from a remote island community (adapted from [25]). Felipe is a 6-year-old boy who lives in a remote village on Kadavu Island with his mother and maternal grandparents. He has a history of motor and speech delays, feeding difficulties and some challenging behaviours. He loves listening to music.

**Table 1 ijerph-20-06732-t001:** Caregiver interviews.

Caregiver Participant Details		
Location	Suva	4
	Serua	4
	Kadavu	4
Interview participants	Female caregiver only	9
	Both male and female caregivers	3
Ethnicity	I-Taukei	9
	Indo-Fijian	3

**Table 2 ijerph-20-06732-t002:** Child characteristics.

Age	Gender	Condition/s *
2	Female	Down Syndrome
2	Female	Cerebral Palsy
2	Male	Down Syndrome, Club Foot
4	Female	Autism spectrum disorder
4	Male	Cerebral Palsy
4	Male	Cerebral Palsy
4	Female	Speech delay, seizures
5	Male	Blind, cleft lip and palate
5	Male	Global developmental delay, seizures
5	Male	Global developmental delay
6	Male	Cerebral Palsy
13	Male	Cerebral Palsy

* Some conditions listed are suspected diagnoses based on presentation rather than formal diagnoses due to lack of formal assessment services.

**Table 3 ijerph-20-06732-t003:** Key stakeholder interviews.

	Stakeholder Location			Stakeholder Level	
Sector	Suva	Serua	Kadavu	Service Provider	Government
Health	3	2	2	6	1
Education	3	-	-	2	1
Disability	3	-	-	2	1
Social Support	2	1	1	3	1
Total	11	3	3	13	4

## Data Availability

Data sharing is not applicable to this article.

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
