# Peer review of "The Journey to Early Identification and Intervention for Children with Disabilities in Fiji"

_ijerph, 2023, doi:10.3390/ijerph20186732_

Round 1
Reviewer 1 Report
I recommend that the authors add a reflection in the conclusion on the possible use of distance support for communication between professionals and families. In situations where this form of communication is sufficient, the financial and time difficulties would be eliminated. Also, the establishment of self-help parent groups with possible distance communication could contribute to improving the quality of support for families with children with disabilities.
Reviewer 2 Report
Dear authors,
First of all, thank you for submitting the manuscript entitled: “The journey to early identification and intervention for children with disability in Fiji” for consideration for publication in the International Journal of Environmental Research and Public Health. The paper is suitable for this Special Issue “Supporting Children and Adolescents with Disabilities and Their Families”, since it aims to explore the experiences of families of children with disability regarding early identification and intervention in Fiji.
The paper is well written and I find the manuscript interesting and significant in the area.
I just have a few comments to the paper in the current form:
l.2 - Perhaps the title is too broad, it just seems like a report. In my opinion the paper should have a more appealing and specific title.
l. 9. Abstract – it could be interesting to add some of the facilitators as well, instead of merely identify the challenges faced.
l. 107 – this sentence should be revised.
l. 108 – the aim of the study is somewhat different than in the abstract. It seems it just focus on the examination of the experiences of 108 families however, I believe it is also appropriate to refer the other relevant key stakeholders from health, education, disability, and social support sectors that were interviewed.
Moreover, I believe authors should delete the word “developmental” since children diagnose is not only developmental disabilities. Here and along the document (e.g. l. 492).
l. 162 – What is the meaning of “Stakeholder level”?
l. 168 - Translation into English was made by experts or by the research team? How did the content validations was made? Was the study approved by an ethical committee?
Minor editing of English language required.
Reviewer 3 Report
Thanks for the opportunity to review this interesting exploratory study! I think the topical focus is probably the best feature of this manuscript, with the general organization and analyses being sufficiently strong. In fact, I do believe this manuscripts meets most of the expectations for IJERPH. However, there are some very basic issues I was a little surprised to see, and I don't think the paper should be published without addressing them. First, authors periodically make assertions that make me a little uncomfortable. Sure, we all know that early intervention is generally a good idea--but, then, we also know that we should be a little less unequivocal in making these assertions (e.g., line 28 on Page 1), even if the assertion is generally true for many children in many contexts. But, as it is currently phrased, the assertion is a dangerous one! There are always exceptions--a great deal, in many cases, and scientific publications should always qualify these blanket assertions. On a similar note, concepts, etc. are sometimes discussed/mentioned, seemingly under the assumption that readers would be familiar with them. In a manuscript of this nature, since not everyone will have the working knowledge of global health-related issues and concepts, let's be sure to briefly introduce them. The authors should make an effort to ensure that they wouldn't catch readers "off guard" with highly specialized concepts, etc. Pls note that I marked "Minor editing of English language required" under "Quality of English Language" for this reason, seeing no other place to indicate this major concern. Yes, the language should be qualified and made less unequivocal, although the author(s) is/are perfectly fluent in the English language! Also, key concepts, etc. should be put into context, so even experts in fields outside of global health would be able to fully appreciate the paper.
Now, the intro section is a bit disappointing in that it essentially brushes over the relevant literature. Upon discussing the general literature on early identification, intervention, etc., let us dwell a bit and talk about the challenges faced in the developing countries. Paragraph two is far too brief and superficial to serve this purpose. Why does that happen? What are some of the mechanisms which lead to such outcomes? Is it about financial and other resources? Again, since this paper would be read by scholars, practitioners, etc. outside of global health, it would be useful for the authors to make sure the paper will make good sense to experts that may not directly be familiar with the notions frequently utilized in global health.
The methodological description leaves much to be desired. "A qualitative study" is a very general way of describing the method. Also, journey maps are useful tools in understanding patient/stakeholder experiences; however, it, in and of itself, is not sufficient in describing the research method here. Why? This is because the mapping can take place via different frameworks of qualitative research methods (e.g., surveys, interviews, focus groups, ethnographic methods, and/or any combination, etc. etc.). For instance, ethnography is a great method to develop journey maps, and is that what transpired in the current study? So, the authors DO need to describe and justify the research method in far more detail and in a far more precise manner.
Along the same vein, given the nature of the data being reported here, I'm very uncomfortable with the label, "results." Findings would be far more appropriate.
Please see above. Grammar, clarity, etc. are generally very good, so the comments here aren't about the command of English, per se. Rather, I am concerned that: (a) unwarranted assertions are made periodically, and they should be made to sound a bit more tentative/less unequivocal; and (b) sometimes, the authors introduce concepts, etc. that are not all that common, as though the readers should be thoroughly familiar with them (e.g., the 2030 Agenda, which shouldn't just be "thrown in" without brief intro on what it is, who is in charge, and what it is intended to accomplish, etc.)
Reviewer 4 Report
This is a report of interviews with and service providers and caregivers of children with disability in Fiji, using journey mapping to identify recommendations for improving services. The Introduction includes a very useful description of the Fijian context. Methods and Results are clear. Discussion highlights major issues and includes limitations. The paper is extremely clearly written. The paper addresses an important topic with practical implications that go beyond Fiji.
A queries and suggestions:
1. Who was on the advisory group? (p 3, line 129) (I don’t mean to ask names, only descriptions of who they were: e.g., caregivers of children with disability, clinicians, researchers, etc.).
2. Were the two interviewers (FS and SP) in any way connected with the Educational system or Disability services in Fiji? In other words, is there any reason to think that participants might have been influenced by an interviewer’s affiliation to give certain responses? And how was that handled?
3. How were caregiver participants identified and recruited?
4. If the journal allows, could the 2 interview guides be attached as appendices or supplementary material?
5. Please provide information about Ethics approval for this study.
6. Under Data Analysis, who was involved in the analysis? Did one person do all, or were there independent raters? Describe that process a little more.
7. Figures 2 to 4 read like individual children, the details are so specific. But the text describes them as “composite” journey maps. If they just 3 examples of the children, then that’s okay (but make it clear that these are pseudonyms). If they are composite, then it doesn’t seem to be appropriate to give such precise details (e.g., name of town, time in NICU, diagnosis), as these will be different for different children.
8. Page 8, line 218: Is “triangulated” the right word here? I understand that term to mean that something is verified from at least two different sources. But the process described here seems to be fitting the comments from the two sets of interviews into the framework, not verifying it.
9. Page 9, lines 255 onwards identify a lack of information at diagnosis stage, as reported by parents. Was there any verification of this information in the interviews with the professionals. What did they say about the diagnosis process? It would be interesting, if you had the data, to know what solutions they see to this problem. In other words, do they agree that insufficient information is given to families at diagnosis? Or do they report something a bit different (e.g., follow-up appointments made but not kept; information given but parent unable to understand it at the time)?
10. Page 9, lines 278 onwards: Similar question. What do the professionals say about this stage? What do they see as the problems and solutions?
11. Page 10, lines 299 onwards: Excellent list of specific recommendations. If families are having trouble coming to the hospital, is there any capacity for staff to do home visits or community visits? If hospital staff cannot manage this, are there community services that could take this work on? (Reading on, I see you’ve raised this possibility at Stage 2.)
